# Giant uniaxial negative thermal expansion in FeZr$_2$ alloy over a wide temperature range

Meng Xu[1], Qiang Li[2], Yuzhu Song[1], Yuanji Xu[3], Andrea Sanson[4,5], Naike Shi[1], Na Wang[1], Qiang Sun [6], Changtian Wang [1], Xin Chen [2], Yongqiang Qiao[6], Feixiang Long[1], Hui Liu [1], Qiang Zhang [7], Alessandro Venier[4], Yang Ren [8], Francesco d'Acapito [9], Luca Olivi [10], Danilo Oliveira De Souza [10], Xianran Xing [2] & Jun Chen [1,11] ✉

Negative thermal expansion (NTE) alloys possess great practical merit as thermal offsets for positive thermal expansion due to its metallic properties. However, achieving a large NTE with a wide temperature range remains a great challenge. Herein, a metallic framework-like material FeZr$_2$ is found to exhibit a giant uniaxial (1D) NTE with a wide temperature range (93-1078 K, $\bar{\alpha}_l = -34.01 \times 10^{-6}\,\mathrm{K}^{-1}$). Such uniaxial NTE is the strongest in all metal-based NTE materials. The direct experimental evidence and DFT calculations reveal that the origin of giant NTE is the couple with phonons, flexible framework-like structure, and soft bonds. Interestingly, the present metallic FeZr$_2$ excites giant 1D NTE mainly driven by high-frequency optical branches. It is unlike the NTE in traditional framework materials, which are generally dominated by low energy acoustic branches. In the present study, a giant uniaxial NTE alloy is reported, and the complex mechanism has been revealed. It is of great significance for understanding the nature of thermal expansion and guiding the regulation of thermal expansion.

Negative thermal expansion (NTE), whose size shrinks on heating, is an unusual physical property. Over the past two decades, the field of NTE has attracted great interest due to its tremendous value in basic research and potential applications to regulate precisely thermal expansion in fuel cells, aerospace instruments, and electronic devices[1-6]. Numerous NTE materials have been discovered in functional materials, and different NTE mechanisms have been deeply studied[7]. Usually, the mechanisms of NTE can be grouped into two main categories, driven by low-frequency phonons or electronic structure transition. For example, the one mainstream NTE

mechanism, the phonon-driven flexible framework NTE material, features a wide NTE temperature window but relatively moderate NTE magnitude, such as ZrW$_2$O$_8$[1], ScF$_3$[8], cyanides[9,10], and MOFs[11]. The other class of NTE resulting from changes of electronic structure endows with a large NTE magnitude but a narrow NTE temperature window, such as spontaneous volume ferroelectrostriction[7], magneto-volume effects (MVE) in magnetic compounds[12], charge transfer[13,14], and Mott transition[15,16].

Designing strong NTE over a wide temperature is the key to the development of NTE. Some novelty methods have been applied to

[1]Beijing Advanced Innovation Center for Materials Genome Engineering, Department of Physical Chemistry, University of Science and Technology Beijing, Beijing 100083, China. [2]Institute of Solid State Chemistry, University of Science and Technology Beijing, Beijing 100083, China. [3]Institute for Applied Physics, University of Science and Technology Beijing, Beijing 100083, China. [4]Department of Physics and Astronomy, University of Padua, Padova I-35131, Italy. [5]Department of Management and Engineering, University of Padua, Padova I-35131, Italy. [6]International Laboratory for Quantum Functional Materials of Henan, School of Physics and Engineering, Zheng-zhou University, Zhengzhou 450001, China. [7]Neutron Scattering Division, Oak Ridge National Laboratory, Oak Ridge, TN 37831, USA. [8]Department of Physics, City University of Hong Kong, Kowloon, Hong Kong, Hong Kong 518057, China. [9]CNR-IOM-OGG c/o European Synchrotron Radiation Facility (ESRF) 71 Av. des Martyrs, 38000 Grenoble, France. [10]ELETTRA Synchrotron Trieste, s.s. 14 km 163,500 in Area Science Park, 34149 Basovizza - Trieste, Italy. [11]Hainan University, Haikou 570228 Hainan Province, China. ✉e-mail: junchen@ustb.edu.cn

achieve giant NTE, such as introducing microstructural effects in the sintered ceramics of Ca$_2$RuO$_4$[15], or reducing the NTE crystallographic directions to lower dimensionality like diyn-diol molecules (large 2D NTE)[17] and Ag$_3$[Co(CN)$_6$] (colossal 1D NTE)[10]. The strong NTE materials over a wide temperature window are predominantly focused on inorganic materials. NTE alloys have broad application potential due to their high electrical/thermal conductivity and good mechanical properties. However, the soft metal bonds in alloys are prone to expansion coupling with the phonons. It is a huge challenge to design NTE alloys with strong NTE over a wide temperature region.

Herein, we have found a giant uniaxial (1D) NTE in the FeZr$_2$ alloy. The FeZr$_2$ ingot exhibits 1D giant NTE from 93 K to 1078 K ($\bar{\alpha}_l = -34.01 \pm 0.02 \times 10^{-6}$ K$^{-1}$), which is the largest NTE ($\Delta l/l_0 = -3.35\%$) with a wide temperature window in all metal-based materials to date. When our work was in progress, the superconductor CoZr$_2$ was reported to show an anomalous thermal expansion, and its uniaxial NTE can be modulated by transition element modifications[18–20]. Here, the FeZr$_2$ alloy complex NTE mechanism is revealed by systematic experimental techniques and first-principles calculations with a comparative study of isostructural alloys of MZr$_2$ (M = Fe and Ni). Interestingly, unlike the conventional framework NTE materials mainly driven by low-frequency phonons, the present giant uniaxial NTE is evoked by the coupling between the weak bond flexible structure and the high-frequency phonons. Moreover, FeZr$_2$ features good metal properties. And it is hysteresis-free during thermal expansion cycling (Supplementary Fig. 1a). The above merits are superior for its practical application. The insightful mechanism also will broaden the NTE families and be instructive for thermal expansion control.

## Results

### Crystal structure and giant 1D NTE

The detailed crystal structures of FeZr$_2$ and the counterpart of NiZr$_2$ were studied by joint NPD and SXRD (see details in Supplementary Figs. 3–4 and Table S1). FeZr$_2$ and NiZr$_2$ were both single same tetragonal phase (space group: $I4/mcm$). M (Fe or Ni) and Zr atoms occupy Wyckoff sites 4a (0, 0, 0.25) and 8 h ($x$, $x$ + 0.5, 0), respectively (Fig. 1a).

It is known that the chemical bonding of intermetallic compounds is complex[21], especially for CuAl$_2$-type materials[22]. Different CuAl$_2$-type materials exhibit notably different bonding behaviors[23–25]. The chemical bonding and the crystal structure of FeZr$_2$ observed from different perspectives are shown in Fig. 1a (details analysis of chemical bonding in Supplementary section 5). It can be found that FeZr$_2$ has a flexible crystal structure connected by the vertex-linked Fe$_2$Zr$_4$ octahedral primitives. The octahedrons Fe$_2$Zr$_4$ are misaligned along the $c$-axis direction, like rows of connected lanterns, exhibiting an open framework-like structure feature. According to the previous reports, Fe–Fe interactions are crucial for FeZr$_2$ to form a CuAl$_2$-type structure[22,25]. Interestingly, in FeZr$_2$, half of the Fe–Fe bond distance (about 1.4 Å) is much larger than the Fe atomic radius (around 1.26 Å)[26], indicating that Fe–Fe bond can stabilize FeZr$_2$ into a CuAl$_2$-type structure with a large $c$-lattice parameter (called large axial ratio $c/a$). The large distance of the Fe–Fe bond implies its abnormal bonding behavior, which will be discussed later.

It is intriguing that FeZr$_2$ ingot exhibits a giant uniaxial NTE ($\Delta l/l_0 = -3.35\%$) with a wide temperature window (93–1078 K) along the vertical direction (Fig. 1d), which is consistent with the trend of the lattice constant of $c$ obtained from neutron pair distribution function (nPDF), neutron powder diffraction (NPD), and synchrotron X-ray diffraction (SXRD) (Fig. 1e). Such excellent linear NTE performance is the largest and widest among all metallic materials to date, such as Hf$_{0.83}$Ta$_{0.13}$Fe$_2$ ($\Delta l/l_0 = -0.17\%$, 222–327 K)[27], La(Fe,Si,Co)$_{13}$ ($\Delta l/l_0 = -0.29\%$, 160–270 K)[28], Er$_2$Fe$_{14}$B ($\Delta a/a_0 \sim -0.27\%$, 10–552 K)[29], Ho$_2$Fe$_{17}$ ($\Delta c/c_0 \sim -0.58\%$, 13–326 K)[30], MnNiGe ($\Delta l/l_0 = -2.37\%$, 80–275 K)[31], SrAu$_3$Ge ($\Delta c/c = -0.29\%$, 110–295 K)[32], CoZr$_2$ ($\Delta c/c \sim -1.19\%$, 7–572 K)[19], and Ca$_{0.85}$La$_{0.15}$Fe$_2$As$_2$ ($\Delta l/l_0 \sim -0.72\%$, 5–300 K)[33]. Moreover, the wide uniaxial NTE temperature region of FeZr$_2$ exceeds that of many non-metallic framework structure NTE materials, such as Ag$_3$[Co(CN)$_6$] ($\Delta c/c = -6.08\%$, 20–496 K)[10], (S,S)-octa-3,5-diyn-2,7-diol ($\Delta c/c = -2.31\%$, 240–330 K)[17], graphite

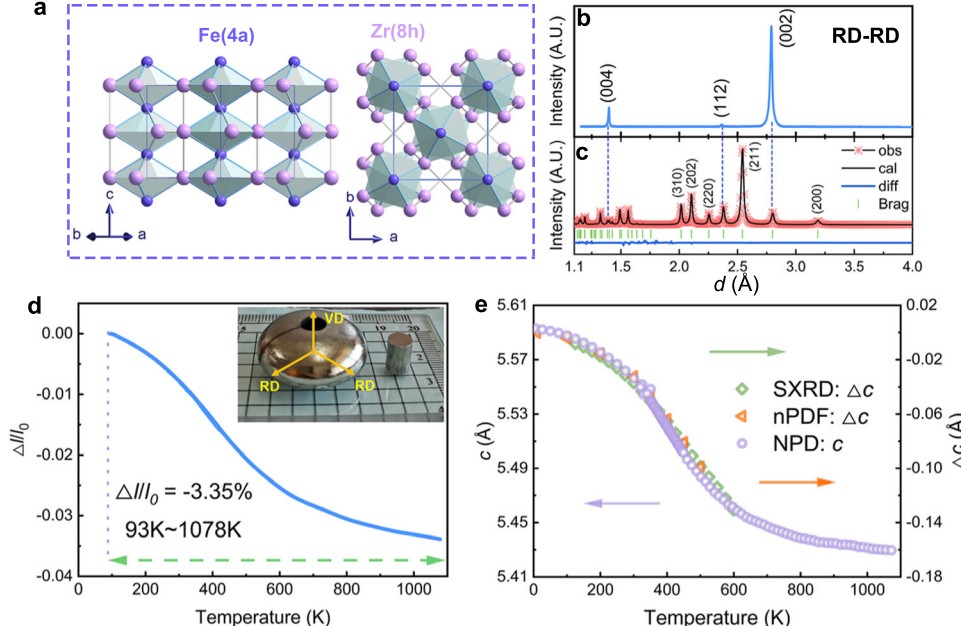

**Fig. 1 | Crystal structure and giant 1D NTE. a** Crystal structure of FeZr$_2$. **b** XRD pattern of the bulk sample FeZr$_2$ measured inside the RD–RD (RD: radial direction) plane. **c** Rietveld refinements of SXRD patterns for FeZr$_2$. **d** Linear thermal expansion measured by a thermo-dilatometer along the vertical direction on heating (93 to 423 K, 300 to 1078 K). **e** Temperature dependence of the lattice parameter $c$ extracted from NPD, and the change of lattice constant $c$, $\triangle c$, extracted by nPDF and SXRD.

($\Delta a/a = -0.16\%$, 200–400 K)[34], In[Au(CN)$_2$]$_3$ ($\Delta c/c = -1.84\%$, 100–395 K)[35], and HMOF-1 ($\Delta b/b = -0.34\%$, 160–320 K)[36].

Comparing the XRD/SXRD patterns of FeZr$_2$ powder and bulk in the RD–RD plane (Fig. 1b, c), it can be found the XRD of the FeZr$_2$ button ingot exhibits a strong texture. There are very few diffraction peaks in RD–RD plane of the bulk (Fig. 1b). Only (002) and (004) peaks possess high intensity. This indicates the giant 1D NTE in FeZr$_2$ ingots is due to the strong texture of the bulk FeZr$_2$ with the crystallographic orientation along [001]//VD melting by arc furnace (details in Supplementary section 1.1). Moreover, the electron backscattered diffraction (EBSD) measurement also shows that FeZr$_2$ ingot has a strong texture with [001]//VD (Supplementary Fig. 9).

In fact, cycling performance is also an important indicator for the practical applications of NTE materials. FeZr$_2$ bulk shows excellent hysteresis free cyclic NTE performance over a wide temperature range (107–567 K) and continuous distribution of the coefficient of thermal expansion (CTE) between −25 and −75 × 10$^{-6}$ K$^{-1}$ (Supplementary Fig. 1a). Moreover, the low-cost, facile, and non-toxic Fe–Zr binary metal-based materials possess natural metallic properties with high electrical/thermal conductivity and good mechanical properties. All these properties in bulk FeZr$_2$ offer an exceptional choice for the practical application of NTE materials.

## Anisotropic thermal vibration

It has been well known that the NTE mechanisms in alloys belong to MVE[37,38], phase transition[39,40], and charge transfer[41,42]. However, the temperature dependence of NPD shows that FeZr$_2$ has no magnetic and phase transition in the measured temperature range (Supplementary Fig. 8). The X-ray absorption near edge structure (XANES) results of the K-edge of Zr and Fe also confirmed there is no electron transfer (Supplementary Fig. 13). Direct evidence excludes these prevalent mechanisms. Interestingly, according to Fig. 1a, it can be observed that FeZr$_2$ has a signature framework feature with a flexible structure and large pore volume. This type of structure could be driven by phonons to produce NTE.

In order to insightfully reveal the mechanism of the giant uniaxial NTE of FeZr$_2$, the thermal expansion of the counterpart NiZr$_2$ and the atomic ADPs for isostructure FeZr$_2$ and NiZr$_2$ systems were studied in detail (see Supplementary for detailed thermal expansion information on the NiZr$_2$ in Fig. 2 and section 1.3). It is very interesting to find that Zr-U12, the most notable differences of ADPs in both Zr and M atoms (Fig. 2a, b), shows a negative value and continued decrease in giant 1D NTE of FeZr$_2$ but an increase in PTE of NiZr$_2$ in the warming process. The opposite variation of Zr-U12 in both materials can be visualized in the schematic diagrams of the thermal vibrations for Zr atoms (Fig. 2c, d). The sign of Zr-U12 determines the long-axis direction of the Zr thermal vibrational ellipsoid in the $ab$-plane. As shown in Fig. 2c, for FeZr$_2$, the Zr atoms in the four corners of the Fe$_2$Zr$_4$ octahedra are more inclined to centrifugal motions. Such motions tend to induce a compression effect of the Fe$_2$Zr$_4$ octahedrons, just like compressing a lantern. In comparison, the Ni$_2$Zr$_4$ octahedra determined by Zr-U12 tend to rotate. The difference between FeZr$_2$ and NiZr$_2$ systems of the anisotropic thermal expansion and their ADPs are in close association, implying the giant 1D NTE in FeZr$_2$ is closely related to the phonon vibrations.

## Phonon analysis

The density functional theory (DFT) calculations can provide insights into the mechanism of giant 1D NTE from phonon vibrations of FeZr$_2$. Figure 3a, b shows the phonon dispersion curves of FeZr$_2$ and NiZr$_2$. Comparing the magnitude of the Grüneisen parameters along the $c$-axis ($\gamma_c$) versus thermal expansion in FeZr$_2$ and NiZr$_2$ (Fig. 3a, b and Supplementary Fig. 6), there is plenty of negative $\gamma_c$ modes for giant 1D NTE FeZr$_2$, while almost all $\gamma_c$ modes are positive for PTE NiZr$_2$. It indicates a large amount of negative $\gamma_c$ driven by anharmonic phonon

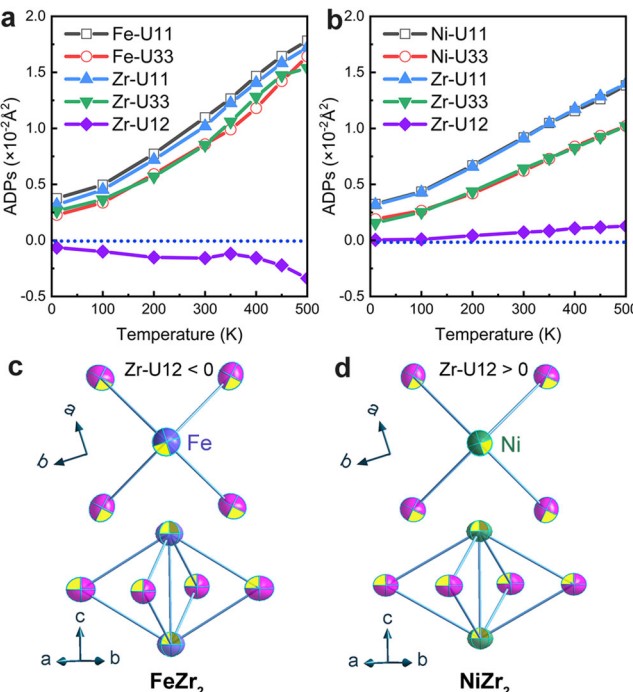

**Fig. 2 | Anisotropic thermal vibration. a, b** Anisotropic displacement parameters (ADPs) of the Zr and M (M = Fe, Ni) atoms in FeZr$_2$ (**a**), and NiZr$_2$ (**b**) extracted by nPDF. **c, d** Thermal ellipsoids along different observation directions for FeZr$_2$ (**c**) and NiZr$_2$ (**d**), respectively.

vibrations is the dynamics factor of giant 1D NTE in FeZr$_2$. Fascinatingly, the negative $\gamma_c$ in FeZr$_2$ generated in the optical region is more multiple and negative than that in the acoustic region (Fig. 3a). It suggests that the distortion caused in the optical phonon modes can induce multiple, large, negative $\gamma_c$, which contributes to a large NTE in the $c$-axis.

The schematics of Zr and Fe representative vibrational modes have been listed in Fig. 3c, in which all modes correspond to the four most negative $\gamma_c$ at the high-symmetry directions. All four models are located in optical branches, representing the distortion of the basic unit Fe$_2$Zr$_4$ octahedra. Three of the four vibrational models (63, 67, and 83 cm$^{-1}$) contribute to the negative value of Zr-U12. It shows that FeZr$_2$ generates multiple, large, negative $\gamma_c$ driven by the high-frequency optical phonons modes, which provides the main contribution to its giant 1D NTE. Moreover, the thermal vibrational ellipsoids of Fe and Zr atoms were almost identical between the experiment extracted from nPDF and the phonon dispersion calculation (Supplementary Fig. 11). The consistent results again verify that the phonon vibration is the cause of NTE. The vibration modes with negative Grüneisen parameter ($\gamma$) contributions to NTE are common in non-metallic framework NTE materials, such as ZrW$_2$O$_8$[43], ScF$_3$[44], and GaFe(CN)$_6$[45]. However, it is rare to find phonon-driven giant (volumetric or uniaxial) NTE in metal-based materials.

It is known that low-frequency phonons dominate NTE in conventional framework materials. NTE in framework materials is gradually weakened at high temperatures since low-frequency phonons vibration modes become unstable. However, in the present framework-like metallic FeZr$_2$, not only do the low-frequency phonon modes generate NTE, but also the high-frequency optical modes excite multiple, large, negative $\gamma_c$ in high temperatures, which further enhances NTE. The high-frequency modes allow FeZr$_2$ to exhibit giant 1D NTE over a wide temperature window.

Furthermore, the distribution of the negative $\gamma_c$ with the vibration frequency can well describe the nonlinear temperature dependence of

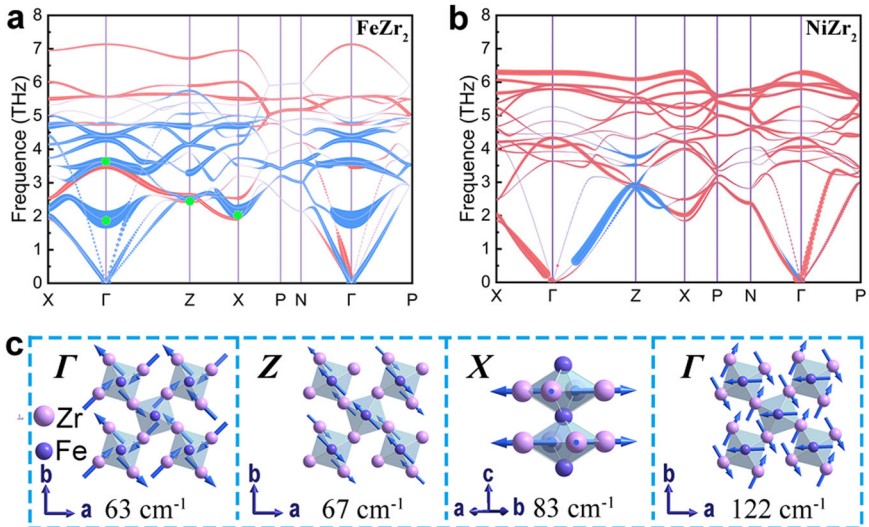

**Fig. 3 | Phonon dispersion curves and the schematics of vibrational modes.**
**a**, **b** Phonon dispersion curves for FeZr$_2$ (**a**) and NiZr$_2$ (**b**), with the size of the dots corresponding to the magnitude of $\gamma_c$; the blue indicates negative $\gamma_c$ and the red indicates positive $\gamma_c$. **c** Representative phonon modes contribute to the four most negative $\gamma_c$ at each high-symmetry point (the green dot in (**a**)) for FeZr$_2$.

the $c$-axis for FeZr$_2$. As shown in Fig. 1e, FeZr$_2$ exhibits a relatively weak NTE at low temperature due to the low-frequency phonons that result in small and negative $\gamma_c$. As the temperature increases, a lot more negative $\gamma_c$ are inspired in the high-frequency optical branches, bringing a larger NTE magnitude. Finally, higher temperature displays an increased proportion of positive $\gamma_c$ leading to a slow dip of NTE.

In metal-based materials, phonons-induced NTE is rare. The phonons-related NTE metal materials have only been found in a few materials, such as Zn[46], Ge[47], As[48], and the intermetallic compound Ca$_{1-x}$La$_x$Fe$_2$As$_2$[33]. However, these NTE materials occur at low temperatures and there is no definitive evidence for an association with high-frequency phonons in metallic materials. FeZr$_2$ is the rare case of the giant uniaxial NTE metallic material with a wide working window, which is dominated by high-frequency optical phonons.

## Electronic structure

To understand the origin of the giant 1D NTE for FeZr$_2$, we need to understand two issues profoundly. One is how FeZr$_2$ with a large Fe–Fe bond can be stabilized as a large axial ratio ($c/a$) CuAl$_2$-type structure, which guarantees large spatial contraction. On the other hand, the essential issue is why the structure of FeZr$_2$ prefers to incur negative $\gamma_c$; especially in the optical phonon region to generate large, negative $\gamma_c$. To answer both questions, we analyzed the chemical bonding as well as the electronic structure for MZr$_2$ (M = Fe and Ni) systems in detail.

The electron density of states (DOS) of the isostructure FeZr$_2$ and NiZr$_2$ illustrates a notable difference in the electronic structure between Fe and Ni. With the increase of electrons in $d$ orbitals of M elements from giant 1D NTE FeZr$_2$ to PTE NiZr$_2$, we find both the DOS of the M atom and Zr atom have considerable value at the Fermi level (E$_f$, Supplementary Fig. 19), which suggests a typical metallic property of these systems. Meanwhile, the DOS of M moves to lower energy from FeZr$_2$ to NiZr$_2$ (Supplementary Fig. 20c, d). A significant difference can be found in the partial DOS (pDOS) of the $dz^2$-orbital of M between −2 and 0 eV. The detailed pDOS of $dz^2$-orbital of M as shown in Fig. 4a, it can be found that the peak of occupied states of Fe is small and located between −2 and 0 eV below the E$_f$. However, it moves away from the E$_f$, and the occupied states become bigger from Fe to Ni. The pDOS change between adjacent atoms will affect the interactions between adjacent atoms[49,50]. Therefore, the pronounced difference in the pDOS around E$_f$ from FeZr$_2$ to NiZr$_2$ will significantly affect the interactions for the M–M and Zr–M.

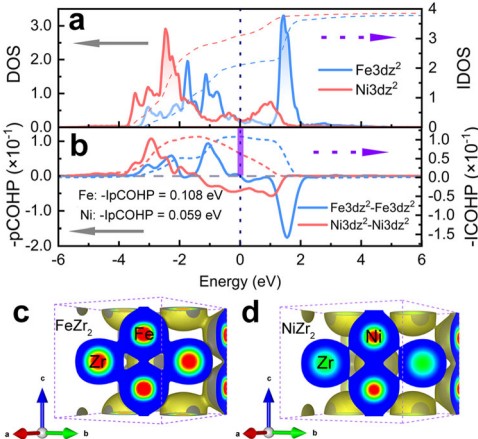

**Fig. 4 | Electron density of states and -COHP. a**, **b** The DOS/IDOS (**a**) of M$dz^2$ (M = Fe, Ni) and -pCOHP/-IpCOHP (**b**) for the most significant change in isostructure of MZr$_2$ (M = Fe, Ni). **c**, **d** Charge distributions of M$_2$Zr$_4$ octahedra in the M–Zr–M–Zr planes with the same isosurfaces at 0.0375 e/r$_0^3$.

The charge density isosurface provides a perspective to visualize the interactions between neighboring atoms. The same charge density isosurface in the MZr$_2$ (M = Fe, Ni) systems was sliced along the Zr–M–Zr plane (Fig. 4c, d). As the atomic number of M increases, the charge density isosurface decreases between Zr and M while it increases significantly between M and M. In general, a large charge overlap between adjacent atoms means strong interatomic interactions[51,52]. The change of charge overlap can qualitatively indicate an increase in the ratio of bond strength (M–M)/(Zr–M) from FeZr$_2$ to NiZr$_2$. It means that a smaller (M–M)/(Zr–M) favors causing bigger $c/a$.

It should be noted that it is not reliable to quantify and directly compare the strength of Zr–Fe and Zr–Ni (or Fe–Fe and Ni–Ni) bonds by the magnitude of the charge density. Because electron density distribution may be either a bonding interaction, or an antibonding interaction that weakens the strength of the chemical bond[50,53]. Subsequently, it will be discussed that the chemical bond strengths of Zr–M and M–M are directly compared by the experimental acquisition of the effective force constants from extended X-ray absorption fine structure (EXAFS) measurement.

**Table 1 | Bond effective force constants of Zr–M and M–M (M = Fe, Ni; $\kappa_{\parallel}$ is the corresponding bond-stretching) extracted from EXAFS[62]; the thermal expansion properties of MZr$_2$ ingot along VD (93–1078 K)**

| Sample | $\kappa_{\parallel}$ (eV/Å$^2$) | | $\triangle l/l_0$ |
|---|---|---|---|
| | Zr–M | M–M | |
| FeZr$_2$ | 2.76 ± 0.03 | 2.36 ± 0.26 | −3.35% |
| NiZr$_2$ | 3.78 ± 0.10 | 3.61 ± 0.28 | 1.12% |

## Discussion

The crystal orbital Hamilton population (-COHP) demonstrates that the interaction of the Fe3$dz^2$-Fe3$dz^2$ plays an essential role in stabilizing flexible large $c/a$ CuAl$_2$ structure in FeZr$_2$. -COHP can provide a detailed chemical bonding analysis between two specified atoms[50,54]. Figure 4b shows the projected -COHP (-pCOHP) of M3$dz^2$-M3$dz^2$ and its corresponding integrated -pCOHP (-IpCOHP: integrated the -pCOHP below E$_f$ represents the magnitude of the interaction between the specified two orbitals[50,54]). From FeZr$_2$ to NiZr$_2$, it can be found that the -pCOHPs of M3$dz^2$-M3$dz^2$ move to the low-energy region (Fig. 4b), and the change of -IpCOHP of M3$dz^2$-M3$dz^2$ was the most significant in all -IpCOHPs of M–M bond (Details of the -IpCHOP in Supplementary section 5.2 and Table S7–S8). Interestingly, the -pCOHP of M3$dz^2$-M3$dz^2$ has many antibonding interactions under E$_f$ in NiZr$_2$. However, below E$_f$, it is almost all bonding interactions in FeZr$_2$. This result in the -IpCOHP of Fe3$dz^2$-Fe3$dz^2$ (0.108 eV) is almost twice as large as that of Ni3$dz^2$-Ni3$dz^2$ (0.059 eV). A larger bonding interaction implies more favorable stability for the chemical bond[50,54]. The considerable bonding interaction of Fe3$dz^2$-Fe3$dz^2$ plays a vital role in Fe–Fe bond (Supplementary Table S8). It is helpful to stabilize the Fe–Fe bond to support the flexible large $c/a$ CuAl$_2$ structure in FeZr$_2$. As a requisite, the long Fe–Fe bonds will provide a large spatial contraction along the $c$-axis.

EXAFS results reveal that the weak bond is the origin of the giant 1D NTE in FeZr$_2$. Table 1 shows that the bond effective force constants of $\kappa_{\parallel}$ of the Zr–M and M–M both decrease from NiZr$_2$ to FeZr$_2$, indicating the Fe$_2$Zr$_4$ octahedron is composed of relatively soft Zr–Fe and Fe–Fe bonds. The magnitude of the chemical bond strength determines the drastically diverse thermal expansion behavior between FeZr$_2$ and NiZr$_2$. As shown in Fig. 5a, b, the "apparent" and "true" thermal expansion of the Zr–Fe bond is much larger than that of the Zr–Ni bond, suggesting that the relatively weak Zr–Fe bond strength provides less constraint to the Fe$_2$Zr$_4$ octahedra. Due to such a large expansion of the Zr–Fe bond, the Fe–Fe bond with relatively weak strength has to contract severely to maintain the Fe$_2$Zr$_4$ octahedron stability. This results in a large shrinkage along the $c$-axis. Meanwhile, the atomic mean-square relative displacements (MSRDs) of Zr–Fe bonds show a larger value both in parallel MSRD$_{\parallel}$ and transverse MSRD$_{\perp}$ than Zr–Ni bonds (Supplementary Fig. 17), suggesting that the Zr atoms in FeZr$_2$ have a larger true displacement. However, the Zr atoms prefer transversely vibrating along the Zr–Zr bonds due to the Zr–Zr bonds constraint (details in Supplementary Fig. 12). As a result, in the optical phonon regions, the vibrational modes correspond to the negative ADPs of Zr-U12, which are prone to occur in FeZr$_2$. In other words, the vibrational modes listed in Fig. 3c (63, 67, and 83 cm$^{-1}$) are more vulnerable to being generated in the optical phonon regions. This contributes to the large NTE by the phonon-driven flexible structure of FeZr$_2$.

Consequently, the large $c/a$ CuAl$_2$-type structure of FeZr$_2$ is stable due to the big Fe3$dz^2$-Fe3$dz^2$ interaction contribution for Fe–Fe bond, which provides a large contraction space along the $c$-direction. The origin of the giant 1D NTE of metallic FeZr$_2$ is the flexible structure driven by phonon vibrations. Especially the multiple, large, negative $\gamma_c$ appears due to coupling the soft bonds in the high-frequency optical

region, which dominates giant 1D NTE with a wide temperature region. The simple model of phonon and bond coupling of FeZr$_2$ is depicted in Fig. 5c. It can be found that the structure change schematic diagram (Fig. 5c) during thermal expansion is similar to wine-rack motion with molecular-based uniaxial NTE materials, such as Zn[Au(CN)$_2$]$_2$[55], methanol monohydrate[56], and Ag$_3$[Co(CN)$_6$][57]. As a comparison, the Ni$_2$Zr$_4$ octahedra consist of relatively strong Zr–Ni and Ni–Ni bonds (Fig. 5d). The stiff bonds have difficulty producing negative $\gamma_c$ driven by the phonons (Fig. 3b), leading to normal expansion in Zr–Ni and Ni–Ni bonds. This causes three crystal dimensional PTE in NiZr$_2$.

In this work, a metal-based giant macro-measurable 1D NTE has been found in FeZr$_2$ due to its strong texture. The origin of giant uniaxial NTE in FeZr$_2$ has been systematically researched by a combined analysis of the temperature dependence of SXRD, NPD, nPDF, EXAFS, and DFT calculations. The strong Fe3$dz^2$-Fe3$dz^2$ interaction contributing to the Fe–Fe bond supports a large $c/a$ CuAl$_2$-type structure, which provides a margin for $c$-axis contraction. What's more, negative $\gamma_c$ is easy to be generated in the flexible framework-like structure of FeZr$_2$, and the soft bonds prefer distortion in the optical region. The large, negative $\gamma_c$ in the optical region dominates the giant 1D NTE over a wide temperature region. The present work reveals the NTE mechanism in the metal-based giant 1D NTE of FeZr$_2$. It provides a direction for new NTE materials design and CTE control.

## Methods

### Synthesis of materials

The ingot samples of MZr$_2$ (M = Fe, Ni) were prepared in a water-cooled copper crucible by arc-melting protected using a high-purity argon atmosphere. All metal constituent elements were used by high-purity elementary substances (purity at least 99.9%). All samples must be turned over and re-melted at least five times to ensure homogeneous composition. Then, all samples were annealed in quartz protected in a high-purity argon atmosphere with 0.06 MPa at 1123 K for 5 days.

### XRD and dependence temperature of SXRD measurements

The room temperature XRD patterns were collected from PANalytical with a Cu target. The XRD patterns of the bulk samples are shown in Fig. 1b. The variable temperature SXRD (125 to 600 K) was measured at the beamline 11-ID-C of APS with the wavelength λ = 0.1173 Å, Argonne National Laboratory, USA.

### Temperature dependence of NPD

The data of the high-resolution variable temperature NPD (10 to 500 K) was collected from the beamline POWGEN of Oak Ridge National Laboratory (ORNL), USA. The high-strength NPD (4 to 350 K, 320 to 1072 K) was obtained at the Wombat beamline of the Australian Nuclear Science and Technology Organisation (ANSTO), Australia. The crystal structure refinements were reined using GASAII software based on the Rietveld method.

### nPDF measurements

Temperature dependence of nPDF for FeZr$_2$ was collected at the beamline POWGEN. Structure refinements over different $r$ values using the CuAl$_2$-type structure (tetragonal phase, $I4/mcm$) model were performed using PDFgui.

### EXAFS measurements

Ni and Zr K-edge EXAFS measurements of NiZr$_2$ were performed at the LISA beamline of ESRF[58], Grenoble (exp. HC-4185), while Fe and Zr K-edge EXAFS measurements of FeZr$_2$ were performed at the XAFS beamline of ELETTRA Synchrotron in Trieste (two different runs were performed for Zr EXAFS of FeZr$_2$, collected during the experiments 20190096 and 20210156). The samples for EXAFS were prepared by mixing and pelletizing the sample powder with boron nitride powder. The amount of sample powder was chosen to have an absorption

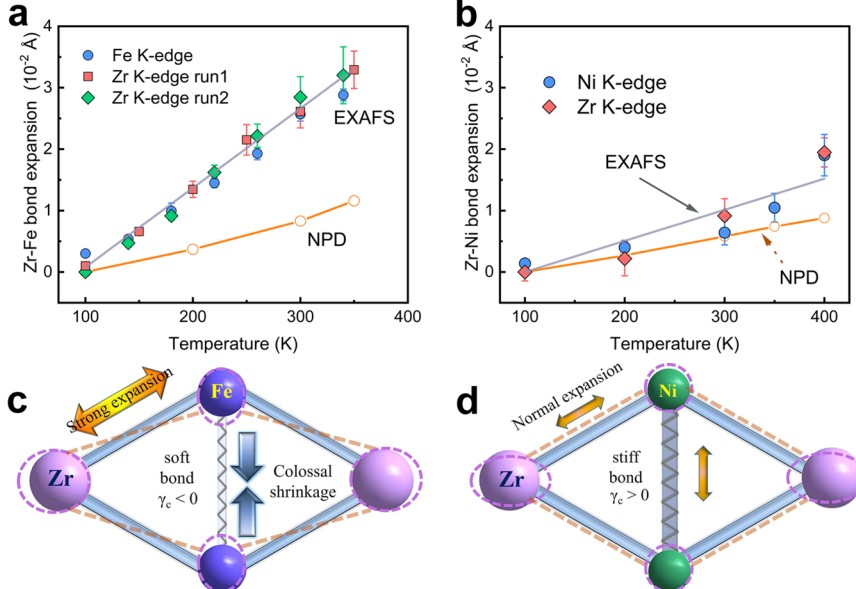

**Fig. 5 | The schematic diagram of the giant 1D NTE. a, b** Zr−M (M = Fe, Ni) bonds expansion: true bond expansion extracted by EXAFS (gray line+symbols) and apparent bond expansion measured by SXRD (orange line+symbols). The bars represent calculated error values. **c, d** 2D simplified geometry, and evolutions for the schematic diagram of giant 1D NTE FeZr$_2$ (**c**) and PTE NiZr$_2$ (**d**).

edge jump Δμx ∼ 1. The EXAFS spectra were collected in transmission mode with an energy step varying from around 0.2 eV (in the near-edge region) to ∼5 eV (at the highest energies) in order to obtain a uniform wave vector step Δk ∼ 0.03 Å$^{-1}$, in the worst case, 0.04 Å$^{-1}$. The X-ray beam was monochromatized by a Si (111) and a Si (311) double crystal monochromator at Fe, Ni K, and Zr K edges, respectively. The samples, kept under high-vacuum conditions (<10$^{-5}$ mbar) during the entire experiment, were mounted in a helium cryostat and the temperature was stabilized and controlled, ensuring thermal stability within ±0.5 K.

**Thermal expansion measurement**

The linear thermal expansion measurement was executed by the thermos-dilatometer (NETZSCH DIL402) with a heating rate of 5 K/min upon heating. The thermal cycling tests use the rate of 3 K/min heating up and cooling down.

**Density functional theory (DFT) calculations**

Our calculations are based on the Vienna ab initio simulation package (VASP) in the framework of density functional theory (DFT). Interactions between ion cores and valence electrons are described by the projector augmented wave[59] method within the Perdew-Burke-Ernzerhof parameterization of the generalized gradient approximation (GGA). The phonon spectrum is calculated using the PHONOPY[60] code. The directional Grüneisen parameter is calculated by

$$\gamma_{i,l} = -\frac{\partial \ln \omega_i}{\partial \ln l}, l = a, c.$$

where $\omega_i$ is the frequency of the $i$th mode, and $l$ is the directional of the supercell.

For the analysis of chemical bonding, we use the crystal orbital Hamilton population (COHP) method[61]. The projected -COHP(-pCOHP) and their energy integrals (-IpCOHP) were calculated using the local-orbital basis suite toward electronic structure (LOBSTER)[54]. Usually, the -ICOHPs/-IpCOHPs of different materials cannot be compared directly because the average electrostatic potentials depend on arbitrary zero energies are possible differences for different systems.

## Data availability

The data that support the findings of this study are available from the corresponding author upon reasonable request.

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

## Acknowledgements

This work was supported by the National Key Research and Development Plan of China (2022YFE0109100), and the National Natural Science Foundation of China (grant nos. 21825102 and 12104038). This research used resources of the Advanced Photon Source, a U.S. Department of Energy (DOE) Office of Science User Facility operated for the DOE Office of Science by Argonne National Laboratory under Contract no. DE-AC02-06CH11357. The authors acknowledge Dr. Chinwei Wang for assisting in collecting the temperature dependence of neutron powder diffraction data using the Wombat high-intensity diffractometer of the Australian Nuclear Science and Technology Organisation (ANSTO).

## Author contributions

M.X. and J.C. find conceived this study and designed the experiments. M.X. carried out the experiments of sample preparation and other measurements. M.X. performed analyzed the data of XRD. Y.Z.S. assisted with the NPD analysis with the help of other authors. M.X., Q.L., and H.L. analyzed the data of nPDF. A.S. performed the EXAFS data collection and analysis. A.V., F.d'A., L.O., and D.O.D.S. assisted with the EXAFS collection and analysis. Y.Q.Q. assisted with the SXRD and XRD analysis. Y.J.X., N.W., Q.S., C.T.W., X.C., and F.X.L. performed DFT calculations. Q.Z. conducted the NPD and nPDF measurements. N.K.S. and Y.R. assisted with the SXRD data collection and analysis. M.X. and J.C. wrote the draft of the paper with contributions from other authors. X.R.X. and J.C. provided useful insights on this project. All authors discussed the results and commented on the manuscript.

## Competing interests

The authors declare no competing interests.
