## [Peer Review File · Nature Communications]

Giant uniaxial negative thermal expansion in FeZr₂ alloy over a wide temperature rangeREVIEWER COMMENTS

Reviewer #1 (Remarks to the Author):

This paper has demonstrated a 1D colossal NTE of FeZr₂ by a lot of experimental evidences. The NTE mechanism has also explained by first-principles calculations. The present results should be interesting and valuable for the developing and application of NTE materials. I have some comments about it. The paper should be accepted for publication after minor modification.

1. The volume of FeZr₂ is PTE as shown in Fig.S4d. The present NTE phenomenon is along the vertical direction or along the solidification direction. The reason given by the authors is the sample with fiber texture due to the preparation by arc melting. The authors should give the morphology of the sample to verify the presume.

2. If the sample with fiber texture, I wonder what would happen when the sample was cut along the planar direction.

3 In the figure S19c of supplemental materials, it should be PDOS of Zr but it was mistyped by Fe.

Reviewer #2 (Remarks to the Author):

Professor Jun Chen and coworkers investigated anisotropic thermal expansion behavior including a uniaxial shrinking of FeZr₂. However, I found many ambiguous and confusing points in the manuscript. Surprisingly, I could not find TE constants of a and b axes. Most importantly, the titled negative behavior is not “colossal”. The colossal thermal expansion has been defined by $\alpha > 100 \text{ MK}^{-1}$, discussed many times by earlier investigation for more flexible systems such as metal-organic frameworks and metal cyanides. Additionally, “ultra-wide temperature range” is not scientific but just strengthen simple 93 K-1078 K. Thus, this manuscript overemphasizes the results in an attempt to make them look better. Uniaxial NTE behavior is not so rare, this is also known from the cited reference papers.

How did the authors calculate the coefficient from non-linear cell constant changes as shown in Fig1d,e? Moreover, the standard deviation should be added into the coefficient values.

I think that -34 MK^{-1} is relatively large, however, the comparison of such linear thermal expansion coefficient value with that of true NTE materials involving volumetric NTE such as ZrW₂O₈, ScF₃, Cd(CN)₂ and MOF-5 is nonsense. The volumetric NTE material is superior to uniaxial NTE materials.

In the discussion part, what does “relatively weak Zr-Fe bond” “relatively weak Fe-Fe bond” mean? I think that Fig4c,d display bond strengths $\text{Zr-Fe} > \text{Zr-Ni}$, thus, these explanations are contradictory.

It is difficult to read discussions (so many place) of this manuscript because there is no proper citation where references should be required.

Therefore, this work is suitable to more specific journal.

Minor comments:

Structural changes during TE are similar to wine-lack motion which has been discussed many times by other molecular based materials. When comparing to such framework materials, ATE and NLC based on the wine-lack topology might be useful for improving this work.

I could not understand negative values of ADP by this text.

I could not catch the connection of between “Those suggest that the anisotropic atomic vibrations of Zr and M atoms are not obvious in both FeZr₂ and NiZr₂” and the context.

Reviewer #3 (Remarks to the Author):

I reviewed the manuscript entitled "Colossal uniaxial negative thermal expansion in FeZr₂ alloy over an ultra-wide temperature range". The paper reports on the observation of uniaxial NTE along c-axis in FeZr₂ and the absence of NTE in NiZr₂. The crystal structures and their temperature evolutions have been examined carefully using NPD, SXRD, and nPDF. In addition, EXAFS, ADP, theoretical investigations on phonon and electronic structure have been done; those results revealed unique optical phonons with high frequency. Since the combined characterizations reported in the paper are convincing and clearly solving the origin of NTE in FeZr₂, I recommend the work to be published in Nature Communications after revisions. The paper will encourage further investigation in the MZr₂ family and related uniaxial (or volume) NTE in new materials. See my comments below for revisions.

1) Uniaxial NTE in MZr₂ has already been reported in recent papers. CoZr₂ and related alloys exhibit uniaxial NTE. The trend in those papers is consistent with the trends reported in this work. Therefore, modification of the introduction part is necessary.

(a) Yoshikazu Mizuguchi, Md. Riad Kasem, Yoichi Ikeda, Anomalous Thermal Expansion in a CuAl₂-type Superconductor CoZr₂, J. Phys. Soc. Jpn. 91, 103601 (2022).

(b) Yuto Watanabe, Hiroto Arima, Hidetomo Usui, Yoshikazu Mizuguchi, Sign change in c-axis thermal expansion and lattice collapse by Ni substitution in Co_{1-x}Ni_xZr₂ superconductors, arXiv:2211.00958.

(c) Hiroto Arima, Md. Riad Kasem, Yoshikazu Mizuguchi, Axis thermal expansion switching in transition-metal zirconides TrZr₂ by tuning the c/a ratio, arXiv:2210.10367.

2) Sample information should be enriched. Is the sample single crystal or poly? Although the sample picture and XRD on the surface are shown, no explanation is there. What is the meaning of HP? How about the homogeneity of the crystal orientation in the case of polycrystalline sample with c-axis orientation.

3) The authors estimated U_{ij} to see the atomic displacements and the relation to NTE. The negative U₁₂ will be one of the causes of c-axis NTE in FeZr₂, but the current explanation is not enough to convince readers. I recommend the authors to add related works on thermal expansion of inorganic materials and ADPs.

4) Abbreviations should be defined at the sentence where that appeared first. Grüneisen parameter γ , HP, XANES, etc. In addition, ADP may be atomic anisotropic displacement parameter [Trueblood et al., Acta Cryst. A52, 770-781 (1996)].

We highly appreciate the referees' comments on our manuscript (Manuscript ID: NCOMMS-22-44514-T). The modified title is "Giant uniaxial negative thermal expansion in FeZr₂ alloy over an ultra-wide temperature range". The comments much improve the quality of the present study. We have revised the manuscript point to point according to the comments. The revised manuscript (Manuscript ID: NCOMMS-22-44514-A) has been resubmitted to *Nature Communications*. The comments are responded to one by one in the following content.

Answers to Reviewer 1

This paper has demonstrated a 1D colossal NTE of FeZr₂ by a lot of experimental evidences. The NTE mechanism has also explained by first-principles calculations. The present results should be interesting and valuable for the developing and application of NTE materials. I have some comments about it. The paper should be accepted for publication after minor modification.

Answers: Thanks very much for the comments. The authors highly appreciate the positive comments for the present study. We have revised the manuscript according to the comments point by point.

(1) The volume of FeZr₂ is PTE as shown in Fig.S4d. The present NTE phenomenon is along the vertical direction or along the solidification direction. The reason given by the authors is the sample with fiber texture due to the preparation by arc melting. The authors should give the morphology of the sample to verify the presume.

Answers: Thanks very much for the comments and suggestions. The morphology of FeZr₂ alloy was examined using SEM with the secondary electron emission model at both low magnification inside the RD-RD plane (Fig. R1a) and high magnification inside the VD-RD plane (Figs. R2a). It can be found that there was no discernible boundary at the grain boundary, indicating that no other phase precipitated at the grain boundary. In the corresponding region of SEM measurement, the electron back-scattering diffraction (EBSD) measurements at RD-RD (Figs R1b,1c) and RD-VD

(Figs R2b,2c) planes both reveal a strong texture in FeZr_2 grains with the orientation of $[001]//\text{VD}$, and the grain orientations of $[010]$ and $[110]$ are randomly distributed in the RD-RD plane. Moreover, the EBSD with the grain boundaries reveals that the grain geometry and size of FeZr_2 are random. To better show the readers the morphology of the FeZr_2 ingot, Fig. R1 has been added to the Supplementary File.

Fig. R1. **a**, The SEM using the secondary electron emission model for FeZr_2 inside the RD-RD plane. **b-c**, The IPF of EBSD patterns along the vertical direction (VD) and the radial direction (RD) of the FeZr_2 ingot. The EBSD samples were prepared using the electropolishing method, which exists in small pits due to the long corrosion time.

Fig. R2. **a**, The SEM for FeZr₂ inside the VD-RD plane. **b-c**, The IPF of EBSD along the vertical direction (VD) and the radial direction (RD) of the FeZr₂.

(2) If the sample with fiber texture, I wonder what would happen when the sample was cut along the planar direction.

Answers: Thanks very much for the comments. Fig. R1a demonstrates the SEM of morphology for FeZr₂ inside the RD-RD plane, which is along the planar direction. The small pits on the surface are caused by the long electropolishing time. In the corresponding region, Fig. R1b-c shows the IPF of the sample in different orientations. It can be found the sample is polycrystalline and has a strong texture in the FeZr₂ phase with [001]/VD. Meanwhile, it can be observed that [010] and [110] are a nearly random distribution of grain orientation in the RD-RD plane.

Fig. R3. shows the linear thermal expansion within the RD-RD plane for FeZr₂ determined by dilatometry along mutually perpendicular directions (RD-X and RD-Y). The ingot exhibits a strong positive thermal expansion (PTE) with the coefficient of thermal expansion (CTE) of $\bar{\alpha}_{l-X} = 30.59 \pm 0.02 \times 10^{-6} \text{ K}^{-1}$ and $\bar{\alpha}_{l-Y} = 29.26 \pm 0.02 \times 10^{-6} \text{ K}^{-1}$ in the RD-X and RD-Y directions between 107 to 500K, respectively. On the other hand, the average CTEs for lattice parameter a are $\alpha_a = 25.75 \pm 0.04 \times 10^{-6} \text{ K}^{-1}$ for FeZr₂ between 10 to 500K ($\alpha_a = 29.75 \pm 0.04 \times 10^{-6} \text{ K}^{-1}$, 100K-500K) extracted from the

NPD (Fig. S4a). Since FeZr_2 is tetragonal crystal symmetry and the EBSD indicates that its grain orientations of [001] and [110] are random distribution inside the RD-RD plane. These result in the planar thermal expansion of the ingot showing nearly the same CTE as the a -axis.

Fig. R3. The linear thermal expansion of FeZr_2 ingot along mutually perpendicular directions (denoted RD-X and RD-Y, respectively) inside the RD-RD plane. The inset shows the sample diagram and the reference coordinate system.

(3) In the figure S19c of supplemental materials, it should be PDOS of Zr but it was mistyped by Fe.

Answers: Thanks very much for the comment and suggestion. We have modified the mistake in the Figure S19 of supplemental materials as the suggestion. And the result of the modification is as follows (Fig. R4).

Fig. R4. The partial density of states of FeZr₂ and NiZr₂.

Answers to Reviewer 2

Comment 1: Professor Jun Chen and coworkers investigated anisotropic thermal expansion behavior including a uniaxial shrinking of FeZr₂. However, I found many ambiguous and confusing points in the manuscript. Surprisingly, I could not find TE constants of *a* and *b* axes.

Answers: Thanks very much for the comments and suggestions. The temperature dependence of NPD shows that the lattice parameter of *a* (*a*=*b*) of FeZr₂ and NiZr₂ both gradually increase with increasing temperature (Fig. R5). The average CTEs of lattice parameter *a* (*a* = *b*) are $\alpha_{\text{Fe},a} = 25.75 \pm 0.04 \times 10^{-6} \text{K}^{-1}$ and $\alpha_{\text{Ni},a} = 10.60 \pm 0.03 \times 10^{-6} \text{K}^{-1}$ for FeZr₂ and NiZr₂ between 10 to 500K, respectively. The corresponding CTEs have been appended to Fig. S4 in the supplementary as required.

Fig. R5. Temperature dependence of lattice parameters *a* for MZr₂ (M = Fe and Ni) extracted from NPD.

Comment 2: Most importantly, the titled negative behavior is not “colossal”. The colossal thermal expansion has been defined by $\alpha > 100 \text{ MK}^{-1}$, discussed many times by earlier investigation for more flexible systems such as metal-organic frameworks and metal cyanides. Additionally, “ultra-wide temperature range” is not scientific but just

strengthen simple 93 K-1078 K. Thus, this manuscript overemphasizes the results in an attempt to make them look better. Uniaxial NTE behavior is not so rare, this is also known from the cited reference papers.

Answers: Thanks very much for the comments and suggestions.

Different types of NTE materials have different potentials for NTE performance. It is a truth that flexible structural systems such as metal-organic frameworks (MOFs) and metal cyanides can exhibit colossal NTE. However, it is difficult to obtain such strong NTE, like MOFs and metal cyanides, in metallic materials. The "colossal" NTE in the previous manuscript is relative to metallic materials. FeZr₂ metal ingot shows a large NTE shrinkage $\Delta l/l_0 = -3.35\%$ (93K~1078K), which is rare and relatively large in metallic NTE materials. In the revision, to be consistent with other types of NTE material comparison standards ("colossal" thermal expansion defined by $|\alpha| > 100\text{MK}^{-1}$), the word "colossal" has been deleted and replaced with "giant" to be consistent with other NTE systems.

To visualize the "ultra-wide temperature range" of FeZr₂, we summarize the operating temperature windows of typical (volumetric or uniaxial) NTE/ZTE metallic materials (Fig. R6). It can be found that FeZr₂ shows the widest NTE working windows in all metallic NTE materials and have more than 300 K wider than the second-ranked Dy₂Fe₁₄B. The quadratically wide NTE temperature range facilitates its application as a thermally compensated material. Therefore, the "ultra-wide" is used to describe the uniaxial NTE temperature range of FeZr₂.

Indeed, uniaxial NTE is not rare. However, it is very rare to achieve a strong and wide temperature range uniaxial NTE in metallic materials. Generally, (volumetric or uniaxial) NTE metallic materials possess a narrow NTE temperature range and low NTE working temperature windows.^{1,2} FeZr₂ alloy exhibits a giant axial NTE and an ultra-wide NTE temperature range ($\Delta l/l_0 = -3.35\%$, 93K~1078K). The axial NTE is strong, and the NTE temperature range is the widest among all metallic NTE materials (Fig. R6). This phenomenon is pretty rare in metallic NTE materials.

Moreover, FeZr₂ alloy shows high hardness with an average Vickers hardness of

419.3Mpa (Fig. R7, Table R1), about twice as high as the classic Invar alloy $\text{Fe}_{0.65}\text{Ni}_{0.35}$ (150~200Mpa).^{3,4} Those indicate FeZr_2 with a large potential application value as a metal thermal compensation material.

Fig. R6. Summary of the (volumetric or uniaxial) NTE/ZTE working temperature windows for typical metallic NTE/ZTE materials⁵⁻²¹

Fig. R7. The morphology of the measured Vickers hardness.

Table R1. The Vickers hardness values per measurement and average Vickers hardness of the FeZr_2 alloy.

H_V (Mpa)						\bar{H}_V (Mpa)
410.7	419.9	429.5	403.4	436.1	416.2	419.3

1. Chen, J., Hu, L., Deng, J. & Xing, X. Negative thermal expansion in functional materials: controllable thermal expansion by chemical modifications. *Chem. Soc. Rev.* **44**, 3522-3567 (2015).
2. Li, Q. *et al.* Chemical diversity for tailoring negative thermal expansion. *Chem. Rev.* **122**, 8438-8486 (2022).
3. Martienssen, W. & Warlimont, H. *Springer handbook of condensed matter and materials data*. Vol. 1 (Springer, 2005).
4. Li, L. *et al.* Good comprehensive performance of Laves phase $\text{Hf}_{1-x}\text{Ta}_x\text{Fe}_2$ as negative thermal expansion materials. *Acta Mater.* **161**, 258-265 (2018).
5. Buschow, K. & Grössinger, R. Spontaneous volume magnetostriction in $\text{R}_2\text{Fe}_{14}\text{B}$ compounds. *Journal of the Less Common Metals* **135**, 39-46 (1987).
6. Zhao, Y. Y. *et al.* Giant negative thermal expansion in bonded MnCoGe -based compounds with Ni_2In -type hexagonal structure. *J. Am. Chem. Soc.* **137**, 1746 (2015).
7. Shen, F. *et al.* Cone-spiral magnetic ordering dominated lattice distortion and giant negative thermal expansion in Fe-doped MnNiGe compounds. *Mater. Horiz.* **7**, 804-810 (2020).
8. Atuchin, V. *et al.* Negative thermal expansion and electronic structure variation of chalcopyrite type LiGaTe_2 . *RSC Adv.* **8**, 9946-9955 (2018).
9. Pandey, A. *et al.* Crystallographic, electronic, thermal, and magnetic properties of single-crystal SrCo_2As_2 . *Phys. Rev. B* **88**, 014526 (2013).
10. Rebello, A., Neumeier, J., Gao, Z., Qi, Y. & Ma, Y. Giant negative thermal expansion in La-doped CaFe_2As_2 . *Phys. Rev. B* **86**, 104303 (2012).
11. Li, W. *et al.* Strong Coupling of Magnetism and Lattice Induces Near-Zero Thermal Expansion over Broad Temperature Windows in $\text{ErFe}_{10}\text{V}_{2-x}\text{Mo}_x$ Compounds. 1009-1015 (2020).
12. Qiao, Y. *et al.* Negative thermal expansion in YbMn_2Ge_2 induced by the dual effect of magnetism and valence transition. *npj Quantum Mater.* **6**, 49 (2021).
13. Andreev, A., De Boer, F., Jacobs, T. & Buschow, K. Thermal expansion anomalies and spontaneous magnetostriction in $\text{R}_2\text{Fe}_{17}\text{C}_x$ intermetallic compounds. *Phys. B* **175**, 361-369 (1991).
14. Algarabel, P., Ibarra, M. J. J. o. m. & materials, m. Invar behaviour and in situ observation of the nitriding process in $\text{R}_2\text{Fe}_{17}\text{N}_x$ intermetallics. **110**, 323-326 (1992).
15. Yanming, H., Liang, F., Zhang, X., Fang, W. & Yanzhao, W. Thermal expansion anomaly and spontaneous magnetostriction of $\text{Gd}_2\text{Fe}_{17}$ compound. *J. Rare Earths* **29**, 772-775 (2011).
16. Wada, H. & Shiga, M. Thermal expansion anomaly and Invar effect of $\text{Mn}_{1-x}\text{Co}_x\text{B}$. *J. Magn. Magn. Mater.* **104**, 1925-1926 (1992).
17. Huang, R. *et al.* Giant negative thermal expansion in NaZn_{13} -type $\text{La}(\text{Fe},\text{Si},\text{Co})_{13}$ compounds. *J. Am. Chem. Soc.* **135**, 11469 (2013).
18. Rohrkamp, J. *et al.* Thermal expansion of the magnetically ordering intermetallics RTMg (R= Eu, Gd and T= Ag, Au). *J. Phys.: Condens. Matter* **19**, 486204 (2007).
19. Takenaka, K. & Takagi, H. Giant negative thermal expansion in Ge-doped anti-

- perovskite manganese nitrides. *Appl. Phys. Lett.* **87**, 261902 (2005).
20. Cao, Y. *et al.* Ultrawide Temperature Range Super-Invar Behavior of $R_2(\text{Fe},\text{Co})_{17}$ Materials (R = Rare Earth). *Phys. Rev. L* **127**, 055501 (2021).
21. Li, W. *et al.* A Seawater - Corrosion - Resistant and Isotropic Zero Thermal Expansion (Zr,Ta)(Fe,Co)₂ Alloy. *Adv. Mater.* **34**, 2109592 (2022).

Comment 3: How did the authors calculate the coefficient from non-linear cell constant changes as shown in Fig1d,e? Moreover, the standard deviation should be added into the coefficient values.

Answers: Thanks very much for the comments and suggestions. Usually, in metallic NTE materials, more attention is paid to macroscopic NTE properties. Here, the linear thermal expansion curve measured by the thermal dilatometer is used in the manuscript. Since the change in thermal expansion of FeZr₂ is non-linear, obtaining a specific CTE value is difficult. Therefore, the average CTE is obtained by a simplified method. The methods are as follows. The standard deviation of the linear thermal expansion has been added as you requested ($\bar{\alpha}_l = -34.01 \pm 0.02 \times 10^{-6} \text{ K}^{-1}$).

The CTE of the linear expansion measured by the thermal dilatometer is calculated as follows:

The calculation of CTE for linear thermal expansion is determined by equation $\alpha_l = \Delta l / \Delta T l_0$.^{22,23} Δl is the variation in the length of FeZr₂ metal block between the highest and lowest temperatures. ΔT is the difference between the lowest test temperature of 93K and the highest test temperature of 1078K. l_0 is the length of the sample at 300K.

The CTEs from non-linear cell constant measured by NPD are calculated as follows:

The CTEs of the a , c , and V are described in the Fig. S4 of the Supplementary File, in conjunction with Comment 1. For the non-linear CTE of the cell constant, it was calculated according to the formula $\alpha_a = (\Delta a/a_0)/\Delta T$.^{22,24} (taking the a -axis as an example, the CTEs are calculated consistently for the c and V). Δa is the difference of lattice parameter a between 10 K and 500 K. a_0 is the lattice parameter a at 10K. ΔT is the given change in temperature. Note that the error values in Fig. S4 are too small and are blocked by the symbol signs. The average CTEs with standard deviation are $\alpha_{\text{Fe},a} =$

$25.75 \pm 0.04 \times 10^{-6} \text{K}^{-1}$ and $\alpha_{\text{Ni},a} = 10.60 \pm 0.03 \times 10^{-6} \text{K}^{-1}$ for lattice parameter a ; $\alpha_{\text{Fe},c} = -33.47 \pm 0.05 \times 10^{-6} \text{K}^{-1}$ and $\alpha_{\text{Ni},c} = 3.27 \pm 0.04 \times 10^{-6} \text{K}^{-1}$ for lattice parameter c ; $\alpha_{\text{Fe},V} = 17.73 \pm 0.07 \times 10^{-6} \text{K}^{-1}$ and $\alpha_{\text{Ni},V} = 24.60 \pm 0.04 \times 10^{-6} \text{K}^{-1}$ for lattice parameter V between 10 to 500K, as shown in Fig. S4a, respectively.

22. Dove, M. T. & Fang, H. Negative thermal expansion and associated anomalous physical properties: review of the lattice dynamics theoretical foundation. *Rep Prog Phys* **79**, 066503 (2016).
23. Atfield, J. P. Mechanisms and Materials for NTE. *Front. Chem.* **6** (2018).
24. Evans, J., Mary, T. & Sleight, A. Negative thermal expansion materials. *Phys. B* **241**, 311-316 (1997).

Comment 4: I think that -34MK^{-1} is relatively large, however, the comparison of such linear thermal expansion coefficient value with that of true NTE materials involving volumetric NTE such as ZrW_2O_8 , ScF_3 , $\text{Cd}(\text{CN})_2$ and MOF-5 is nonsense. The volumetric NTE material is superior to uniaxial NTE materials.

Answers: Thanks very much for the comments and suggestions. It is true that the comparison of linear thermal expansion coefficient with cubic non-metallic framework NTE materials is nonsense. In the revision, some typical uniaxial non-metallic framework NTE materials are selected to provide an intuitive comparison of the uniaxial NTE performance for FeZr_2 . The corresponding part of the manuscript is revised as follows:

“Moreover, the ultra-wide uniaxial NTE temperature region of FeZr_2 exceeds that of many non-metallic framework structure NTE materials, such as $\text{Ag}_3[\text{Co}(\text{CN})_6]$ ($\Delta c/c = -6.08\%$, 20-496K),²⁵ (*S,S*)-octa-3,5-diyne-2,7-diol ($\Delta c/c = -2.31\%$, 240-330K),²⁶ graphite ($\Delta a/a = -0.16\%$, 200-400K),²⁷ $\text{In}[\text{Au}(\text{CN})_2]_3$ ($\Delta c/c = -1.84\%$, 100-395K),²⁸ and HMOF-1 ($\Delta b/b = -0.34\%$, 160-320K).²⁹”

Generally, volumetric NTE material is superior to uniaxial NTE materials. However, for metal uniaxial NTE materials, strong texture samples can be generated by arc melting. This allows its metallic bulk to exhibit NTE properties at a macroscopic scale in a certain direction (usually in the strong texture direction). This also allows tuning of uniaxial NTE metallic materials to achieve axial ZTE/LTE materials with high mechanical performance, which shows promising applications. For example, high

mechanical uniaxial ZTE/LTE (low thermal expansion) dual-phase alloys can be achieved by designing the chemical composition in uniaxial NTE metallic materials.^{4,30,31} FeZr₂ exhibits a giant and wide uniaxial NTE. A high mechanical performance and broad temperature range uniaxial LTE alloy had been achieved in Fe-Zr binary alloy by tuning the phase content of FeZr₂. The LTE alloy has outstanding mechanical performance and the widest LTE temperature range among all metallic LTE/ZTE materials (Figs. R8 and R9). It shows FeZr₂ with great prospects for applications in structural function materials.^{1,30}

Fig. R8. a. The linear thermal expansion along the VD for S1 ingot of Fe-Zr dual-phase alloy. **b.** Comparison of ZTE/LTE properties in metallic-based materials.^{4,30-44} S1 is a dual-phase alloy of FeZr₂ and Zr. The uniaxial thermal expansion regulation by the strong texture of FeZr₂ and its phase ratio. (unpublished work)

Fig. R9. a. Compressive stress-strain curves along the VD for S1 ingot of Fe-Zr dual-phase alloy. **b.** A review of compressive strain versus ultimate strength in the metallic ZTE/LTE materials.^{4,30,34,37,39,40,43-46} (unpublished work)

25. Goodwin, A. L. *et al.* Colossal positive and negative thermal expansion in the framework material $\text{Ag}_3[\text{Co}(\text{CN})_6]$. *Science* **319**, 794-797 (2008).
26. Das, D., Jacobs, T. & Barbour, L. Exceptionally large positive and negative anisotropic thermal expansion of an organic crystalline material. *Nat. Mater.* **9**, 36-39 (2010).
27. Yoon, D., Son, Y.-W. & Cheong, H. Negative thermal expansion coefficient of graphene measured by Raman spectroscopy. *Nano letters* **11**, 3227-3231 (2011).
28. Korcok, J. L., Katz, M. J. & Leznoff, D. B. Impact of metallophilicity on “colossal” positive and negative thermal expansion in a series of isostructural dicyanometallate coordination polymers. *J. Am. Chem. Soc.* **131**, 4866-4871 (2009).
29. DeVries, L. D., Barron, P. M., Hurley, E. P., Hu, C. & Choe, W. “Nanoscale lattice fence” in a metal–organic framework: interplay between hinged topology and highly anisotropic thermal response. *J. Am. Chem. Soc.* **133**, 14848-14851 (2011).
30. Yu, C. *et al.* Plastic and low-cost axial zero thermal expansion alloy by a natural dual-phase composite. *Nat. Commun.* **12**, 1-8 (2021).
31. Lin, K. *et al.* High performance and low thermal expansion in Er-Fe-V-Mo dual-phase alloys. *Acta Mater.* **198**, 271-280 (2020).
32. van Schilfgaarde, M., Abrikosov, I. & Johansson, B. Origin of the Invar effect in iron–nickel alloys. *Nature* **400**, 46-49 (1999).
33. Nakamura, Y., Takenaka, K., Kishimoto, A. & Takagi, H. Mechanical Properties of Metallic Perovskite $\text{Mn}_3\text{Cu}_{0.5}\text{Ge}_{0.5}\text{N}$: High-Stiffness Isotropic Negative Thermal Expansion Material. *J. Am. Chem. Soc.* **92**, 2999-3003 (2009).
34. Zhao, Y.-Y. *et al.* Giant negative thermal expansion in bonded MnCoGe -based compounds with Ni_2In -type hexagonal structure. *J. Am. Chem. Soc.* **137**, 1746-1749 (2015).
35. Dan, S., Mukherjee, S., Mazumdar, C. & Ranganathan, R. Zero thermal expansion with high Curie temperature in $\text{Ho}_2\text{Fe}_{16}\text{Cr}$ alloy. *RSC Advances* **6**, 94809-94814 (2016).
36. Li, S. *et al.* Zero thermal expansion achieved by an electrolytic hydriding method in $\text{La}(\text{Fe},\text{Si})_{13}$ compounds. *Advanced Functional Materials* **27**, 1604195 (2017).
37. Liu, J. *et al.* Realization of zero thermal expansion in $\text{La}(\text{Fe},\text{Si})_{13}$ -based system with high mechanical stability. *Mater. Des.* **148**, 71-77 (2018).
38. Song, Y. *et al.* Zero thermal expansion in magnetic and metallic $\text{Tb}(\text{Co},\text{Fe})_2$ intermetallic compounds. *J. Am. Chem. Soc.* **140**, 602-605 (2018).
39. Song, Y. *et al.* Opposite thermal expansion in isostructural noncollinear antiferromagnetic compounds of Mn_3A (A= Ge and Sn). *Chem. Mater.* **30**, 6236-6241 (2018).
40. Wang, J. *et al.* Balancing negative and positive thermal expansion effect in dual-phase $\text{La}(\text{Fe},\text{Si})_{13}/\alpha\text{-Fe}$ in-situ composite with improved compressive strength. *J. Alloys Compd.* **769**, 233-238 (2018).
41. Hu, J. *et al.* Adjustable magnetic phase transition inducing unusual zero thermal expansion in cubic RCo_2 -based intermetallic compounds (R= Rare Earth). *Inorg. Chem.* **58**, 5401-5405 (2019).

42. Yuan, X. *et al.* Design of negative/nearly zero thermal expansion behavior over a wide temperature range by multi-phase composite. *Mater. Des.* **203**, 109591 (2021).
43. Pang, X. *et al.* Design of zero thermal expansion and high thermal conductivity in machinable xLFCS/Cu metal matrix composites. *Composites, Part B* **238**, 109883 (2022).
44. Zhou, H. *et al.* Low-melting metal bonded MM'X/In composite with largely enhanced mechanical property and anisotropic negative thermal expansion. *Acta Mater.* **229**, 117830 (2022).
45. Predki, W., Knopik, A. & Bauer, B. Engineering applications of NiTi shape memory alloys. *Materials Science Engineering: A* **481**, 598-601 (2008).
46. Lin, K. *et al.* High performance and low thermal expansion in Er-Fe-V-Mo dual-phase alloys. *Acta Mater.* **198**, 271-280 (2020).

Comment 5: In the discussion part, what does “relatively weak Zr-Fe bond” “relatively weak Fe-Fe bond” mean? I think that Fig4c,d display bond strengths Zr-Fe>Zr-Ni, thus, these explanations are contradictory.

Answers: Thanks very much for the comments and suggestions. In the manuscript, we would like to compare the relative strength of the chemical bonds. So the correct expression should be “relatively weak Zr-Fe bond strength” and “relatively weak Fe-Fe bond strength”. Thank you very much for pointing this out, we have revised it in the manuscript.

Fig. 4c-d are not intended for a direct comparison of bond strengths, but rather serve to qualitatively determine that there is a link between the chemical bond strength ratio (M-M)/(Zr-M) and the axis ratios (c/a). As shown in Fig. 4c,d, as the atomic number of M increases, the charge density isosurface decreases between Zr and M while it increases significantly between M and M. In general, a large charge overlap between adjacent atoms means strong interatomic interactions.^{47,48} It can qualitatively obtain the magnitude of the chemical bond strengths is Zr-Fe>Zr-Ni and Fe-Fe<Ni-Ni. Based on these, the magnitude of the chemical bond strength ratio can be obtained $\frac{\text{Fe-Fe}}{\text{Zr-Fe}} < \frac{\text{Ni-Ni}}{\text{Zr-Ni}}$ (the derivation process is as follows). **This result is consistent with the one calculated by the effective force constant extracted by EXAFS**

$$\left(\frac{\text{Fe-Fe}}{\text{Zr-Fe}} = \frac{2.36}{2.76} \approx 0.86, \frac{\text{Ni-Ni}}{\text{Zr-Ni}} = \frac{3.61}{3.78} \approx 0.96, \text{ the error value is not considered to simplify} \right.$$

the calculation). For MZr_2 ($M = Fe, Co, \text{ and } Ni$) system in other work, we found that the c -axis NTE is closely related to the axis ratio (c/a) and the chemical bond strength ratio ($M-M/Zr-M$). Hence, “a small $(M-M)/(Zr-M)$ favors causing big c/a ” is proposed here.

However, it is not reliable to quantify and directly compare the strength of Zr-Fe and Zr-Ni (or Fe-Fe and Ni-Ni) bonds by the magnitude of the charge overlap. Because electron density distribution may be either a bonding interaction, or an antibonding interaction that weakens the strength of the chemical bond.^{49,50} For example, the charge density of Ni-Ni is found to be much larger than that of Zr-Ni in $NiZr_2$ (Fig. 4d). But the actual EXAFS measured results show that the strength of the Zr-Ni bond is stronger than the Ni-Ni bond. Therefore, to compare the chemical bond strength, the effective force constants extracted from EXAFS are applied for comparison. The result shows that the bond strengths $Zr-Fe < Zr-Ni$.

Considering this may mislead the readers and to better reveal the uniaxial NTE mechanism. The manuscript's corresponding part has been revised. The derivation process for the ratio of chemical bond strengths:

$$\therefore Zr-Fe > Zr-Ni, Fe-Fe < Ni-Ni$$

$$\therefore \frac{Zr-Fe}{Zr-Ni} > 1, \frac{Fe-Fe}{Ni-Ni} < 1$$

$$\therefore \frac{Zr-Fe}{Zr-Ni} > 1 > \frac{Fe-Fe}{Ni-Ni}$$

$$\therefore \frac{Fe-Fe}{Zr-Fe} < \frac{Ni-Ni}{Zr-Ni}$$

47. Liu, Y. *et al.* Negative thermal expansion in isostructural cubic ReO_3 and ScF_3 : A comparative study. *Comput. Mater. Sci.* **107**, 157-162 (2015).
48. Hu, L. *et al.* Atomic Linkage Flexibility Tuned Isotropic Negative, Zero, and Positive Thermal Expansion in $MZrF_6$ ($M = Ca, Mn, Fe, Co, Ni, \text{ and } Zn$). *J. Am. Chem. Soc.* **138**, 14530-14533 (2016).
49. Mulliken, R. Electronic population analysis on LCAO-MO molecular wave functions. IV. Bonding and antibonding in LCAO and valence-bond theories. *J Chem Phys* **23**, 2343-2346 (1955).
50. Steinberg, S. & Dronskowski, R. The crystal orbital Hamilton population (COHP) method as a tool to visualize and analyze chemical bonding in intermetallic compounds. *Crystals* **8**, 225 (2018).

Comment 6: It is difficult to read discussions (so many place) of this manuscript because there is no proper citation where references should be required.

Answers: Thank you very much for your comments and suggestions. In the revision, we have made reasonable modifications to the discussion and related sections. And relevant references are cited in appropriate places in the manuscript for more detailed understanding by the readers. In addition, detailed graphs have been added in the supplementary. The modified sections are marked using bright colors in the revision.

Comment 7: Structural changes during TE are similar to wine-lack motion which has been discussed many times by other molecular based materials. When comparing to such framework materials, ATE and NLC based on the wine-lack topology might be useful for improving this work.

Answers: Thanks very much for the comments and suggestions. We speculate that ATE and NLC are abbreviations of anisotropic thermal expansion and negative linear compressibility, respectively. The structure change schematic diagram (Fig. 5c) during thermal expansion is similar to wine-rack motion with molecular based NTE materials, such as $\text{Zn}[\text{Au}(\text{CN})_2]_2$,⁵¹ methanol monohydrate,⁵² and $\text{Ag}_3[\text{Co}(\text{CN})_6]$ ⁵³. Moreover, the uniaxial NTE materials with wine-rack topology are a strong predictor to find NLC property.^{51,54,55} FeZr_2 alloy has a wine-rack-like structure (Fig 1.a-b). It means that FeZr_2 alloy might possess NLC behavior. Therefore, the citation of literatures about ATE and NLC based on the wine-rack topology in the manuscript will improve this work. It will help readers to better understand the evolution of the structure and attract more potential readers. The modifications were implemented in the manuscript as follows:

“It can be found that the structure change schematic diagram (Fig. 5c) during thermal expansion is similar to wine-rack motion with molecular based uniaxial NTE materials, such as $\text{Zn}[\text{Au}(\text{CN})_2]_2$,⁵¹ methanol monohydrate,⁵² and $\text{Ag}_3[\text{Co}(\text{CN})_6]$ ⁵³.”

51. Cairns, A. B. *et al.* Giant negative linear compressibility in zinc dicyanoaurate.

Nat. Mater. **12**, 212-216 (2013).

52. Fortes, A. D., Suard, E. & Knight, K. S. Negative linear compressibility and massive anisotropic thermal expansion in methanol monohydrate. *Science* **331**, 742-746 (2011).
53. Goodwin, A. L., Keen, D. A. & Tucker, M. G. Large negative linear compressibility of $\text{Ag}_3[\text{Co}(\text{CN})_6]$. *Proc. Natl. Acad. Sci.* **105**, 18708-18713 (2008).

Comment 8: I could not understand negative values of ADP by this text.

Answers: Thanks very much for the comment. The atomic displacement parameters (ADPs) express the thermal vibrations of atoms in a crystal. It describes the average displacement of an atom from its average position due to thermal motion.^{56,57} In isotropic crystals, the U_{ii} ($i = 1, 2, \text{ and } 3$) value is a scalar quantity indicating the intensity of the thermal vibrations of the atoms. In anisotropic crystals, ADPs are typically represented by a 3×3 symmetric matrix, and U_{ii} and U_{ij} ($i \neq j, i/j = 1, 2, \text{ and } 3$) are characterized the anisotropy of the intensity and direction of thermal vibrations of atoms. The U_{ii} represents the mean-square displacement of an atom along each of the three orthogonal crystallographic axes. And the U_{ii} is usually a positive value. The U_{ij} represents the correlation between the atomic displacements along different orthogonal crystallographic axes. The U_{ij} has a positive or negative sign. When the U_{ij} value is positive, the vibration direction of the atom is consistent with the positive direction in the crystal structure. And when the U_{ij} value is negative, the vibration direction of the atom is opposite to the positive direction.^{57,58}

Take FeZr_2 as an example, three independent ADPs ($U_{11}=U_{22}$, U_{33} , and U_{12}) can be used to represent the thermal vibrational ellipsoids of Zr according to its crystal environment. In order to better explain the relationship between the negative value of Zr- U_{12} and the giant uniaxial NTE of FeZr_2 , the thermal vibrational ellipsoids of ADPs for Zr atoms at a given value are shown in Fig. R10. It can be found that Zr thermal vibrations can occur when Zr- U_{12} is a negative value (Fig. R10c). In addition, Zr- U_{12} exhibits a tendency for Zr to vibrate along the Zr-Fe bond when it is a negative value (this is described in a simplified way because there is a small angle between the vibrational orientation and the Zr-Fe bond). Fig. 2a,b shows that the biggest difference in the ADPs between FeZr_2 (giant uniaxial NTE) and NiZr_2 (PTE) is Zr- U_{12} . The ADPs

of Zr-U12 is a negative value in FeZr₂ and a positive value in NiZr₂. A link can be found between negative ADPs and NTE, thus guiding us to conduct subsequent studies on phonon spectroscopy calculations, bond strengths, and electronic structures to reveal the mechanism of the giant NTE.

Fig. R10. Schematic diagram of the ADPs of Zr at different values. The yellow arrows denote the tendency of the Zr atoms to vibrate at the set of anisotropic ADPs.

54. Cai, W. & Katrusiak, A. Giant negative linear compression positively coupled to massive thermal expansion in a metal–organic framework. *Nat. Commun.* **5**, 4337 (2014).
55. Cairns, A. B. & Goodwin, A. L. Negative linear compressibility. *Physical Chemistry Chemical Physics* **17**, 20449-20465 (2015).
56. Cruickshank, D. The analysis of the anisotropic thermal motion of molecules in crystals. *Acta Crystallogr.* **9**, 754-756 (1956).
57. Dunitz, J. D., Schomaker, V. & Trueblood, K. N. Interpretation of atomic displacement parameters from diffraction studies of crystals. *J. Phys. Chem. C* **92**, 856-867 (1988).
58. Trueblood, K. *et al.* Atomic displacement parameter nomenclature. Report of a subcommittee on atomic displacement parameter nomenclature. *Acta Crystallogr., Sect. A: Found. Crystallogr.* **52**, 770-781 (1996).

Comment 9: I could not catch the connection of between “Those suggest that the anisotropic atomic vibrations of Zr and M atoms are not obvious in both FeZr₂ and NiZr₂” and the context.

Answer: Thanks very much for the comment. In the previous manuscript, we would like to describe the phenomenon of the changing trend of U_{ii} in the MZr₂ system. It

was found that the differences between U11 and U33 of Zr and M atoms are small (Fig. 2a,b). However, in non-metallic framework NTE materials, the linked atoms of polyhedra with strong transverse vibrations will produce a large difference between the two Uii, such as ZrW_2O_8 ⁵⁹, $\text{Ln}(\text{CN})_6$ ⁶⁰, ScF_3 ⁶¹, and $\text{Ag}_3[\text{Co}(\text{CN})_6]$ ⁶². Here, we found that describing this phenomenon is not well related to the context and can cause confusion to the readers. Therefore, to avoid confusion for the readers, we have modified and removed unnecessary content without affecting the content of the discussion.

59. Evans, J., Mary, T., Vogt, T., Subramanian, M. & Sleight, A. Negative thermal expansion in ZrW_2O_8 and HfW_2O_8 . *Chem. Mater.* **8**, 2809-2823 (1996).
60. Duyker, S. G., Peterson, V. K., Kearley, G. J., Ramirez-Cuesta, A. J. & Kepert, C. J. Negative thermal expansion in $\text{LnCo}(\text{CN})_6$ (Ln= La, Pr, Sm, Ho, Lu, Y): mechanisms and compositional trends. *Angew. Chem., Int. Ed.* **52**, 5266-5270 (2013).
61. Hu, L. *et al.* New Insights into the Negative Thermal Expansion: Direct Experimental Evidence for the "Guitar-String" Effect in Cubic ScF_3 . *J. Am. Chem. Soc.* **138**, 8320-8323 (2016).
62. Goodwin, A. L. *et al.* Colossal positive and negative thermal expansion in the framework material $\text{Ag}_3[\text{Co}(\text{CN})_6]$. *Science* **319**, 794-797 (2008).

Answers to Reviewer 3

I reviewed the manuscript entitled “Colossal uniaxial negative thermal expansion in FeZr₂ alloy over an ultra-wide temperature rang”. The paper reports on the observation of uniaxial NTE along *c*-axis in FeZr₂ and the absence of NTE in NiZr₂. The crystal structures and their temperature evolutions have been examined carefully using NPD, SXRD, and nPDF. In addition, EXAFS, ADP, theoretical investigations on phonon and electronic structure have been done; those results revealed unique optical phonons with high frequency. Since the combined characterizations reported in the paper are convincing and clearly solving the origin of NTE in FeZr₂, I recommend the work to be published in Nature Communications after revisions. The paper will encourage further investigation in the MZr₂ family and related uniaxial (or volume) NTE in new materials. See my comments below for revisions.

Answers: Thanks very much for the comments. The authors highly appreciate the positive comments for the present study. We have revised the manuscript according to the comments point by point.

1) Uniaxial NTE in MZr₂ has already been reported in recent papers. CoZr₂ and related alloys exhibit uniaxial NTE. The trend in those papers is consistent with the trends reported in this work. Therefore, modification of the introduction part is necessary.

(a) Yoshikazu Mizuguchi, Md. Riad Kasem, Yoichi Ikeda, Anomalous Thermal Expansion in a CuAl₂-type Superconductor CoZr₂, J. Phys. Soc. Jpn. 91, 103601 (2022).

(b) Yuto Watanabe, Hiroto Arima, Hidetomo Usui, Yoshikazu Mizuguchi, Sign change in *c*-axis thermal expansion and lattice collapse by Ni substitution in Co_{1-x}Ni_xZr₂ superconductors, arXiv:2211.00958.

(c) Hiroto Arima, Md. Riad Kasem, Yoshikazu Mizuguchi, Axis thermal expansion switching in transition-metal zirconides TrZr₂ by tuning the *c/a* ratio, arXiv:2210.10367.

Answers: Thank you very much for the comments and suggestions. And thank you very much for recommending the three papers, which we did not pay attention to when preparing the manuscript for submission. In the revision, these three important papers have been cited in the introduction part. The modifications in the introduction part are as follows:

“When our work was in progress, the superconductor CoZr_2 was reported to show an anomalous thermal expansion, and its uniaxial NTE can be modulated by transition element modifications.⁶³⁻⁶⁵”

In fact, we found the uniaxial NTE property of FeZr_2 in December 2018. After a systematic study in 2019, a preliminary understanding of the uniaxial thermal expansion properties of MZr_2 (M=Fe, Co, and Ni) and related systems as well as the mechanism have been established. After 2020, we have submitted numerous proposals for further study of the uniaxial NTE mechanism through NPD and SXRD beamtime with the subject of uniaxial NTE of MZr_2 (M=Fe, Co, and Ni). Three of the proposals are the proposal (ID: P9333) at Wombat beamline of the Australian Nuclear Science and Technology Organisation (ANSTO), the proposals (IPTS-26252.1) at POWGEN and (IPTS-26899.1) at NOMAD of the Oak Ridge National Laboratory (ORNL). Below is a screenshot of the proposal systems (Fig. R11).

After nearly four years of research, we have thoroughly studied the uniaxial NTE mechanism and thermal expansion regulation of MZr_2 systems. Many interesting phenomena have been found, such as achieving high mechanical properties uniaxial ZTE over a wide temperature region. These works we are going to submit immediately after this work. In addition, I also finished defending my PhD in December 2022 with the MZr_2 systems as the research object.

a ACNS Neutron Beam Instrument Proposal

ID P9333
 Title Colossal Positive and Negative Thermal Expansion in the Non-Magnetic Metal $MZr_2(M=Fe,Co)$
 Round 2021-1 Neutron
 Type Normal
 Status Unsuccessful
 Date Created 15/9/2020
 Date Submitted 15/9/2020

Mandatory Requirements (not met)

Safety Mandatory Requirement: SDSs must be provided before commencing experiment. If commercial SDSs are not available, completed SDS compilation forms must be provided. The proposal cannot proceed until all mandatory requirements have been fulfilled.

Researchers

Name	Role	Attending
Jun Chen (Uni. Sci. Tech. Beijing)	Principal Scientist	Yes
Meng Xu (Uni. Sci. Tech. Beijing)	Co-proposer (editor)	Yes
Chin-Wei Wang (NSRRC)	Co-proposer	Yes

b Proposal IPTS-26252.1 Type: General User

Title: Colossal Positive and Negative Thermal Expansion in the Non-Magnetic Metal $MZr_2(M=Fe,Co)$
 PI: Jun Chen
 PI Employer: University of Science and Technology Beijing

Abstract

The common phenomenon of positive thermal expansion (PTE) brings a critical issue especially for the applications of aerospace, high precise instruments, and electronic devices. After two-decade studies on the topic of negative/zero thermal expansion (NTE/ZTE), there has been a great progress in aspects of exploring new NTE materials, controlling of thermal expansion, and illuminating NTE mechanisms. [1-5] Up to now, several NTE materials have been discovered in functional materials, such as ferroelectrics, magnetics, multiferroics, and superconductors. ZTE is one of the most important issues for the studies of NTE. However, there are few ZTE materials, such as Invar alloys, [6] which can be applied in reality.

It would be very meaningful to find a wide temperature range NTE material in non-magnetic alloys. Recently, we have found that $FeZr_2$ exhibits positive and negative thermal expansion an order of magnitude greater than that seen in other alloy materials. This alloy material expands along one set of directions at a rate comparable to the most weakly bound solids known. By flexing like lattice fencing, the framework couples this to a contraction along a perpendicular direction. Moreover, no work has reported its NTE property and investigated the relationship between its structure transition and NTE property elaborately. Part substitution of Fe by Ni in $FeZr_2$, or Co by Ni in $CoZr_2$ could change the thermal expansion property. We have tuned it from NTE to ZTE and to PTE successfully by Ni substitution in $CoZr_2$. The mechanisms of controlled NTE properties in MZr_2 system are vague. Based on our previous study on pure $FeZr_2$, it should be strongly related to the transverse vibrations of Zr atoms. We need to investigate the temperature-dependence information Fe atoms details by neutron powder diffraction (NPD), while X-ray diffraction is not sensitive to displacement parameters of Zr atoms, which is important to understand the nature of NTE.

Publications:

Facility: SNS Run Cycle: SNS 2021-A Instrument Scientist Contacted: No
 Instrument(s) BL-11A - POWGEN 3 day(s) req. Total: 3
 BL-11A - Do You Also Want Synchrotron x-ray data? No

c Proposal IPTS-26899.1 Type: General User

Title Colossal Positive and Negative Thermal Expansion in the Intermetallic $MZr_2(M=Fe, Co, Ni)$
 PI: Jun Chen PI Employer: University of Science and Technology

Abstract

The common phenomenon of positive thermal expansion (PTE) brings a critical issue, especially for aerospace applications, high precision instruments, and electronic devices. After two-decade studies on negative/zero thermal expansion (NTE/ZTE), there has been great progress in exploring new NTE materials, controlling thermal expansion, and illuminating NTE mechanisms. [1-5] Up to now, several NTE materials have been discovered in functional materials, such as ferroelectrics, magnetics, multiferroics, and superconductors. ZTE is one of the most important issues for the studies of NTE. However, few ZTE materials, such as Invar alloys [6], can be applied in reality.

Recently, we have found that $FeZr_2$ exhibits positive and negative thermal expansion orders of magnitude greater than that seen in other alloy materials. This alloy material expands along with one set of directions at a rate comparable to the most weakly bound solids known. By flexing like lattice fencing, the framework couples this to a contraction along a perpendicular direction. Moreover, no work has reported its NTE property and investigated the relationship between its local crystalline structure transition and NTE property elaborately. Part substitution of Fe by Ni in $FeZr_2$ or Co by Ni in $CoZr_2$ could change the thermal expansion property. We have tuned it from NTE to ZTE and to PTE successfully by Ni substitution in $CoZr_2$. The mechanisms of controlled NTE properties in the MZr_2 system are vague. Based on our previous study on pure $FeZr_2$, the mechanism should be strongly related to local crystalline structure transition due to Co/Fe substituted by Ni. We need to investigate the temperature-dependence distance change information of Fe-Ni, Co-Ni, Fe-Fe, Fe-Co, and Co-Co pairs details by Neutron pair-distribution-function (NPDF). At the same time, X-ray diffraction couldn't distinguish Fe, Co, Ni, which is vital to understand the nature of NTE to measure NPDF.

Previous HFIR/SNS Publications

Facility: SNS Run Cycle: SNS 2021-B Instrument Scientist Contacted: No
 Instrument BL-1B - NOMAD 3 day(s) req. Total: 3

Fig. R11. Submitted proposals on MZr_2 ($M=Fe, Co, \text{ and } Ni$) system.

63. Arima, H., Kasem, M. R. & Mizuguchi, Y. Axis thermal expansion switching in transition-metal zirconides $TrZr_2$ by tuning the c/a ratio. *Appl. Phys. Express* (2022).
64. Mizuguchi, Y., Kasem, M. R. & Ikeda, Y. Anomalous thermal expansion in a

CuAl₂-type superconductor CoZr₂. *J. Phys. Soc. Jpn.* **91**, 103601 (2022).

65. Watanabe, Y., Arima, H., Usui, H. & Mizuguchi, Y. Sign change in *c*-axis thermal expansion and lattice collapse by Ni substitution in Co_{1-x}Ni_xZr₂ superconductors. *arXiv preprint arXiv:2009.00958* (2022).

2) Sample information should be enriched. Is the sample single crystal or poly? Although the sample picture and XRD on the surface are shown, no explanation is there. What is the meaning of HP? How about the homogeneity of the crystal orientation in the case of polycrystalline sample with *c*-axis orientation.

Answers: Thank you very much for the comments. EBSD measurements reveal that the sample is polycrystalline (Fig. R1.b-c). The grain size (50μm~200μm) of the sample is relatively large (Fig. R1.b-c and Fig. R2.b-c), which is due to the sample being prepared using arc melted and annealed at high temperature for five days. EBSD reveals that the polycrystalline FeZr₂ existence of strong texture with [001]//VD (Fig. R1b), and [110] and [001] grains have a nearly random orientation inside the RD-RD plane (Fig. R1c). HP denotes the horizontal plane of the ingot in the planar stationary state. To avoid confusing the readers, we use RD-RD instead of HP in the revised manuscript.

Fig. R1. **a**, The SEM using the secondary electron emission model for FeZr₂ inside the RD-RD plane. **b-c**, The IPF of electron back-scattering diffraction (EBSD) patterns along the vertical direction (VD) and the radial direction (RD) of the FeZr₂ ingot. The

EBSD samples were prepared using the electropolishing method, which exists in small pits due to the long corrosion time.

Fig. R2. **a**, The SEM for FeZr₂ inside the VD-RD plane. **b-c**, The IPF of EBSD along the vertical direction (VD) and the radial direction (RD) of the FeZr₂.

3) The authors estimated U_{ij} to see the atomic displacements and the relation to NTE. The negative U_{12} will be one of the causes of c -axis NTE in FeZr₂, but the current explanation is not enough to convince readers. I recommend the authors to add related works on thermal expansion of inorganic materials and ADPs.

Answers: Thanks very much for the comments and suggestions.

It indeed helps the reader better understand the mechanism of the c -axis NTE for FeZr₂ to add related works on the thermal expansion of inorganic materials and ADPs. The U_{ii} ($i = 1, 2,$ and 3) and U_{ij} ($i \neq j$) (ADPs) are determined by the vibrational modes of the atoms. In FeZr₂, the ADP of Zr- U_{12} is negative due to the dominance of the negative γ_c vibrational modes. Some inorganic materials due to ADPs induced NTE have been added to the manuscript, making it easier for readers to understand the relationship between ADPs and c -axis NTE in FeZr₂. Add content in the manuscript as follows: “The vibration modes with negative Grüneisen parameter (γ) contributions to

NTE are common in non-metallic framework NTE materials, such as ZrW_2O_8 ⁵⁹, ScF_3 ⁶⁶, and $\text{GaFe}(\text{CN})_6$ ⁶⁷. However, it is rare to find phonon-driven giant (volumetric or uniaxial) NTE in metal-based materials.”

Moreover, It can be found that the variable temperature ADPs, measured at another beamline in ORNL (Fig. R12), are consistent with those of the POWGEN beamline. The ADPs of Zr-U12 is decreasing in FeZr_2 , while that of NiZr_2 is creasing with increasing temperature. The consistent experimental results indicate the reliability of the experimental data. On the other hand, it can be found that the variable temperature U_{ij} exhibits negative values in MOF of NTE materials (the U_{ij} is extracted from the cif of the appendix).^{68,69} Interestingly, some U_{ij} , such as C8-U12 and In1-U23, show a regular decrease in the 3D NTE MIL-68(In) (Fig. R13).⁶⁹

Fig. R12. a-b ADPs of the Zr and M (M = Fe, Ni) atoms in FeZr_2 (a), and NiZr_2 (b) extracted by nPDF from the NOMAD beamline at ORNL. The nPDF data of MZr_2 (M = Fe, Co, and Ni; a total of five components) collected at NOMAD are the next work to explain the fine tuning of the MZr_2 thermal expansion. (Unpublished work)

Fig. R13. a-c, Temperature dependent lattice parameters extracted from Le bail fits of SXRD and fitted lines with second-order polynomial. d, Temperature dependent changes of In1-U23 and C8-U12.^{68,69}

66. Li, C. W. *et al.* Structural relationship between negative thermal expansion and quartic anharmonicity of cubic ScF_3 . *Phys. Rev. L* **107**, 195504 (2011).
67. Gao, Q. *et al.* Low-frequency phonon driven negative thermal expansion in cubic $\text{GaFe}(\text{CN})_6$ Prussian blue analogues. *Inorg. Chem.* **57**, 10918-10924 (2018).
68. Chen, L. *et al.* Thermal enhancement of luminescence for negative Thermal Expansion in Molecular Materials. *Journal of the American Chemical Society* **144**, 13688-13695 (2022).
69. Liu, Z. *et al.* 3D negative thermal expansion in orthorhombic MIL-68 (In). *Chem. Commun.* **54**, 5712-5715 (2018).

4) Abbreviations should be defined at the sentence where that appeared first. Gruneisen parameter γ , HP, XANES, etc. In addition, ADP may be atomic anisotropic displacement parameter [Trueblood *et al.*, *Acta Cryst.* A52, 770-781 (1996)].

Answers: Thank you very much for the comments and suggestions. We have corrected

the corresponding abbreviations in the original text where they first appeared, as you requested.

REVIEWERS' COMMENTS

Reviewer #1 (Remarks to the Author):

I have no other comments. It should be accepted for publication.

Reviewer #2 (Remarks to the Author):

I checked the revised manuscript carefully. I think that most of the ambiguous points seems to have been resolved. Although I do not like the overstatement word, ultra-wide temperature range, I could understand the widest area for the negative thermal expansion of the alloy than other alloys. Such high functionalities will be suitable to the publication by Nature communications.

I would like to point out a minor point finally, the order of figures does not fit the text flow, for example, the first figure is "Supplementary Fig. 9a" in the introduction part. Please modify them via rearrangements of figures for easy reading.

Reviewer #3 (Remarks to the Author):

This is the second round of review of this manuscript.

I think that the authors have addressed all issues. The results are new and should be of great interest for wide-scope readers.

I recommend publication of this paper in Nature Communications.

We highly appreciate the referees' comments on our manuscript (Manuscript ID: NCOMMS-22-44514-A). The modified title is "Giant uniaxial negative thermal expansion in FeZr₂ alloy over a wide temperature range". The comments much improve the quality of the present study. We have revised the manuscript point to point according to the comments. The revised manuscript (Manuscript ID: NCOMMS-22-44514-B) has been resubmitted to *Nature Communications*. The comments are responded to one by one in the following content.

Reviewer #2 (Remarks to the Author):

Comment 1: I checked the revised manuscript carefully. I think that most of the ambiguous points seems to have been resolved. Although I do not like the overstatement word, ultra-wide temperature range, I could understand the widest area for the negative thermal expansion of the alloy than other alloys. Such high functionalities will be suitable to the publication by Nature communications.

Answers: Thanks very much for the comment and suggestion. The authors highly appreciate the positive comments for the present study. The 'ultra-wide' is replaced with 'wide' as required.

Comment 2: I would like to point out a minor point finally, the order of figures does not fit the text flow, for example, the first figure is "Supplementary Fig. 9a" in the introduction part. Please modify them via rearrangements of figures for easy reading.

Answers: Thanks very much for the comments and suggestions. The order of Supplementary Fig. 9a has been exchanged as required in the revised manuscript, and the order of figures has been changed to fit the text flow in the revised manuscript and Supplementary file.

Thank you very much for considering the manuscript, and please feel free to contact us if any additional information is required.

Best wishes

Jun Chen